# Ectomycorrhizal Community Shifts at a Former Uranium Mining Site

**DOI:** 10.3390/jof9040483

**Published:** 2023-04-18

**Authors:** Olga Bogdanova, Erika Kothe, Katrin Krause

**Affiliations:** Microbial Communication, Institute of Microbiology, Friedrich Schiller University Jena, D-07743 Jena, Germany

**Keywords:** ectomycorrhiza, morphotype, metal, plant, field, pot, inoculation

## Abstract

Ectomycorrhizal communities at young oak, pine, and birch stands in a former uranium mining site showed a low diversity of morphotypes with a preference for contact and short-distance exploration strategies formed by the fungi *Russulaceae*, *Inocybaceae*, *Cortinariaceae*, *Thelephoraceae*, *Rhizopogonaceae*, *Tricholomataceae*, as well as abundant *Meliniomyces bicolor*. In order to have better control over abiotic conditions, we established pot experiments with re-potted trees taken from the sites of direct investigation. This more standardized cultivation resulted in a lower diversity and decreased prominence of *M. bicolor*. In addition, the exploration strategies shifted to include long-distance exploration types. To mimic secondary succession with a high prevalence of fungal propagules present in the soil, inoculation of re-potted trees observed under standardized conditions for two years was used. The super-inoculation increased the effect of lower abundance and diversity of morphotypes. The contact morphotypes correlated with high Al, Cu, Fe, Sr, and U soil contents, the dark-colored short-distance exploration type did not show a specific preference for soil characteristics, and the medium fringe type with rhizomorphs on oaks correlated with total nitrogen. Thus, we could demonstrate that field trees, in a species-dependent manner, selected for ectomycorrhizal fungi with exploration types are likely to improve the plant’s tolerance to specific abiotic conditions.

## 1. Introduction

Post-mining areas constitute severe disturbances of the ecosystem with considerable soil damage. Waste rock heaps, their footprint, or open pit surfaces are generally low in nutrients, depleted of organic matter, and might contain high amounts of toxic metals due to acid mine drainage (AMD [1]). At the same time, the substrate is characterized by low porosity, decreased water-holding capacity, and the lack of water-stable aggregates due to compaction. As a result of the low content of clay-humic complexes and acidic pH, metals are almost not absorbed or sedimented, which facilitates their migration and transfer into food chains and, therefore, represents a threat to public health. Within the European Union, former mining areas are required to be rehabilitated to reduce severe effects on the environment and human health [2].

An example of such remediation activities is found at the former uranium mining site near Ronneburg, Germany. After the removal of the waste rock material, a topsoil cover was applied to stop further oxidative processes and to re-establish vegetation, including re-forestation in primary succession. The extreme fluctuations of abiotic conditions with the heterogeneous distribution of toxic metals and acidity, combined with low nutrient and soil moisture, resulted in locally successful tree growth, leaving large spots of topsoil unvegetated. Therefore, the sustainability and stability of land cover need improvement. Remediation methods developed for such sites might become especially helpful in the future when heavy rain events or drought spells become more prominent with climate change.

One measure that may reduce stress on planted trees is the mutually beneficial association with ectomycorrhizal fungi, mostly belonging to the *Basidiomycota*. *Betulaceae*, *Pinaceae*, and *Fagaceae* are prominent trees in temperate forest ecosystems and provide promising candidates for such approaches [3,4,5]. The mutually beneficial symbiosis, in addition to the general traits of nutrient exchange, alleviates abiotic stress like drought or can protect from bioavailable heavy metals, as well as increasing resistance against pathogens [5,6,7]. The positive effects on tree growth and metabolism can raise the survival of plants during afforestation [8,9,10].

The invasion of short roots by the ectomycorrhizal fungus leads to establishing the symbiotic morphotypes with a hyphal mantle surrounding the root and the Hartig’ net providing an extended surface for the exchange of signal molecules as well as water and nutrients between the root cells of the tree, leaving the vascular tissue untouched [11]. Differentiation and distribution of extraradical mycelium reflect their functional traits and describe how fungi explore and interact with the environment defining fungal exploration types [12,13]. Here, different fungal species can be found to form specific exploration types, e.g., the short-contact type of *Inocybe* species [14]. While in nutrient-rich environments with amino acids, ammonium, and nitrate available, contact and short-distance exploration type fungi are generally found to be more prominent, scarce, and spatially dispersed and/or insoluble nutrient supplies lead to increased formation of medium and long-distance rhizomorphs that can transport nutrients towards the host tree over larger distances [15,16,17].

Like with plant communities, succession types for ectomycorrhiza can be defined, and these are distributed into “early-stage” or “late-stage” mycorrhizal interactions [18]. While *Inocybe*, *Laccaria*, and *Thelephora* are dominant at early successional stages, *Lactarius*, *Cortinarius*, *Russula*, or *Tricholoma* are typical for later stages of succession in different abiotic conditions of primary succession [19,20,21]. 

Fungi occurring in metal-contaminated soils are exposed to selective environmental pressure leading to adaption by developing particular metal resistance mechanisms [22,23]. Often these mechanisms are based on a modification of metal adaptation that causes decreased or increased metal mobility, e.g., by the release of organic acids [24]. It has been shown that some ectomycorrhizal fungal species demonstrate a high tolerance to environmental stress, including metal toxicity and low pH, and, therefore, can be applied in soil remediation programs [25,26]. Inoculation of plants with ectomycorrhizal fungi alone or together with associated bacteria might either intensify phytoextraction by plants and lead to the accumulation of heavy metals in aboveground biomass [27,28] or immobilize and sequester metal ions in fungal biomass by biosorption to chitin or melanin, and bioaccumulation with the transport of metals inside the cells by different compounds, e.g., metallothioneins, and surrounding soil limiting plant metal uptake [22,29,30,31]. While for metal removal in phytoextraction, often trees of the species *Salix* or *Populus* have been planted, a valid and sustainable land-use strategy might involve afforestation of the very large, heterogeneously and low-level contaminated lands with temperate forest climax species trees like oak, beech, birch, maple, or coniferous pine and spruce that can be harvested and at the same time provide recreational space without harmful dispersal of metals in the environment. In that case, phytostabilization is the process providing the highest potential. Since fungi can determine plant metal uptake, a stable mycorrhization should be reached, albeit with a high-stress impact, especially on young trees.

With this study, we combined direct observation of the roots of young trees in a former mining site with a set-up using re-potted trees from that same area in a pot experiment. This allowed us to create comparable abiotic conditions. In addition, the pot experiments could be inoculated in addition to the already established mycorrhiza to see whether the presence of additional, late-stage mycorrhiza propagules would improve tree health in a metal-contaminated substrate from a former mining heap site. Thus, this additional experiment was used to mimic secondary succession in a short time frame. The results may be useful to establish re-forestation programs after the closure of mine operations in AMD-impacted post-mining scenarios. Species and functional diversity of mycorrhiza were studied to gain insight into the establishment of mutual symbiosis under detrimental environmental conditions.

## 2. Materials and Methods

### 2.1. Experimental Settings

The field site, Kanigsberg, at the former uranium mining area near Ronneburg, Germany, is characterized by a high content of mobile metals, low pH, and high structural heterogeneity. Naturally occurring stands of young birch (*Betula* sp.), oak (*Quercus* sp.), and pine (*Pinus* sp.) trees, all approximately 2 to 3 years of age, were analyzed. Visually undamaged trees of approximately the same height were sampled: birches of 25–30 cm, oaks of 7–10 cm, and pines of 7–11 cm (Table 1). The topsoil layer was carefully removed to expose the root system, and several soil aggregates directly attached to the roots (along the whole root system when possible) were carefully sampled (rhizosphere soil; three replicates per tree species). Soil not directly connected to tree roots was taken for comparison (bulk soil; Appendix A). Native roots were sampled for mycorrhiza characterization.

At the same site of the former uranium mining site, soil and plant material were retrieved to set up a pot experiment. This was established with soil sampled from Kanigsberg (50°49′35.27″ N, 12°9′12.13″ E) in a greenhouse experiment to control abiotic conditions. The pre-sieved control pot substrate was dried for three days, sieved (<2.8 mm), and filled into Mitscherlich pots (12 L; Appendix A). To prevent direct contact with the soil substrate and pot metal walls, pots were inlaid with polyethylene film. Naturally occurring trees were sampled for the pot experiment, including 15 × 15 cm soil cores from the above-mentioned field site. Soil cores were carefully removed before planting with two trees of one species per pot with 6 kg of a substrate prepared as mentioned above. For additional mycorrhizal inoculation, a blend of mycorrhizal fungi in peat substrate containing *Amanita muscaria*, *Boletus edulis*, *Hebeloma crustuliniforme*, *Paxillus involutus*, *Pisolithus tinctorius*, *Cenococcum geophilum*, *Pisolithus arrhenius* and the endomycorrhizal fungus *Rhizophagus irregularis* was used (INOQ Forst, Schnega, Germany). The inoculation was performed according to the manufacturer’s recommendations. All pots, including control variants, were watered with deionized water to approximately 15 volume % water content. Watering was performed weekly in the autumn-winter seasons and twice per week in the spring-summer seasons. To improve air exchange, the topsoil crust was carefully broken with non-metal equipment the day after watering. The experimental variants (Appendix A) contained at least three replicates of pots resulting in 10 to 14 plant replicates for each tree species. After two years, rhizosphere soil and roots were sampled.

### 2.2. Soil Chemical Analysis

To perform chemical analysis, the soil was air-dried at 40 °C, crushed, and sieved (<2.0 mm). Soil pH was measured with 5 g of pre-treated soil mixed with 25 mL of deionized water shaken for 1 h (pH electrode SenTix 81; Xylem Analytics, Weilheim, Germany) with three technical replicates. Elemental analysis was performed with the air-dried soil, finely ground in an agate mortar stored in 50 mL air-tight falcon tubes in darkness. Contents of total carbon (TC) and total nitrogen (TN) were measured with a VarioMAX CN Element Analyzer (Elementar Analysensysteme, Hanau, Germany). The contents were calculated corresponding to the absolute dry weight (measured at 105 °C). Total phosphorus (TP; ICP-AES, PerkinElmer, Waltham, MA, USA) and the bioavailable fractions of toxic metals (ICP-OES, 725ES, Agilent, Waldbronn, Germany; Quadrupole-ICP-MS (XSeries II, Thermo Scientific, Bremen, Germany) were determined after sequential extraction [32]. The method used has been devised specifically for former mining site substrates. In our investigation, specifically, the two fractions containing bioavailable elements in the mobile fraction (F1), extracted with 1 M NH_4_NO_3_, and specifically adsorbed fraction (F2), extracted using 1 M NH_4_OAc at pH 6,0, were considered. Standard reference material SPS-SW2 (Spectrapure Standards AS) and NIST 1643e (NIST) were analyzed, and the multielement standard solution (500 mg/L Ca, K, Mg, Bernd Kraft) was measured and compared to the certified values to prove the measurement accuracy. For every variant of the experiment, element analyses were performed in triplicates.

### 2.3. Morphotyping

To analyze ectomycorrhizal fungal communities, five trees of each species were sampled (20 × 20 cm soil cores, 10–15 cm depth, ca. 50–100 cm apart from each other). To perform morphotyping, the roots were soaked overnight in tap water at 4 °C, carefully washed on a sieve, and cut into 1–3 cm pieces before the fine roots were separated and observed with a dissecting microscope (Stemi, 2000-C, Zeiss, Jena, Germany).

To analyze the mycorrhizosphere of pot plants, the soil was carefully removed from pots in portions to prevent disturbance, and the tree was removed and immediately processed as described above. Root systems of three plants of each tree species, non-inoculated and inoculated separately, were observed.

The morphology was described according to Agerer [33], with color, type of ramification, the shape of unramified ends, mantle surface, characteristics of rhizomorph, and characteristics of emanating hyphae (compare Appendix A). The abundance of each morphotype per total root length was assessed. Mycorrhizal root tips of different morphology were collected for molecular identification.

### 2.4. Molecular Identification

The small sample size of individual mycorrhizal root tips combined with a substrate rich in heavy metals that may hinder polymerase amplification and the presence of iron and manganese hydroxides that unspecifically bind nucleic acids warranted special care to be taken for DNA sequencing. Root sections from selected short roots were placed in 1.5 mL microcentrifuge tubes containing 0.5 mL sterile distilled water and vortexed for 30 s to remove soil particles. Roots were then transferred to 1.5 mL microcentrifuge tubes with 200 µL 30% H_2_O_2_ and vortexed for 10 s for surface sterilizing, rinsed in sterile distilled water three times, and transferred into a sterile 1.5 mL microcentrifuge tube kept at −20 °C until further processing. Direct PCR [34] using primers specific for fungal internal transcribed spacer (ITS) was performed then on a small piece of the hyphal mantle taken under the binocular. This piece was placed in 20 µL of sterile distilled water and taken for PCR amplification. This led to the successful identification of some of the morphotypes.

When the direct PCR approach was not successful, DNA was extracted from sampled morphotypes. The different individual samples were first extracted using PowerSoil DNA Isolation Kit (MoBio, Carlsbad, CA, USA) after milling with sterile glass beads (Sigma Aldrich, Taufkirchen, Germany) using a plastic pestle, vortexing with 100 µL of sterile distilled water for 30 s and transfer of the supernatant. Bovine serum albumin (Carl Roth, Karlsruhe, Germany) was added at a final concentration of 0.4 µg/µL to reduce the inhibition of *Taq* polymerase. After mixing by pipetting, in a final 50 µL volume, the PCR was performed with 6.25 µL 10x Dream *Taq* Buffer, 1.25 µL 10 mM dNTPs, 5 µL 10 µM each forward (ITS1: 5′-TTCGTAGGTGAACCTGCGG-3′) and reverse (ITS4: 5′-TCCTCCGCTTATTGATATGC-3′) primers [35], and 1.25 U *Taq* polymerase. After pre-heating for 5 min at 95 °C, 35 cycles (95 °C 30 s, 56 °C 30 s, 72 °C 50 s) and final elongation at 72 °C for 10 min followed. Amplified DNA was visualized and documented after agarose gel electrophoresis using 1 µg/mL ethidium bromide under UV light.

The PCR products were purified (QIAquick, Qiagen, Germany) and sequenced (GATC, Konstanz, Germany) using primer ITS1. Sequences were compared to entries in NCBI (http://www.ncbi.nlm.nih.gov; accessed on 17 March 2023) and UNITE [36] databases.

### 2.5. Data Processing

Ectomycorrhizal (ECM) communities of field trees were characterized based on the relative abundance of morphotypes with community diversity indices (Shannon diversity index (H_SH_)), Gini-Simpson index (H_GS_), Simpson dominance index (H_SD_), Berger–Parker index (H_BP_)). Trees of one species were compared to each other within each variant of the experiment (field trees, non-inoculated pot trees, inoculated pot trees) to determine the similarity/dissimilarity of ECM communities with Sørensen, Jaccard, and Bray–Curtis indices and to estimate representability of replicates.

Canonical correspondence analysis (CCA) was performed to estimate the correlation of exploration types of ectomycorrhiza and field morphotypes with environmental variables. A *p*-value lower than 0.05 was considered significant. An open-source software, PAST 4.03, was used for all multivariate analyses. The significance of multivariate analysis results was checked with a one-way analysis of similarities (ANOSIM) and one-way permutational analysis of variance (PERMANOVA) and calculated by permutation of group membership (N = 9999). Bonferroni correction was applied. 

Shapiro–Wilks’s test of normality was performed with JASP 0.14.0.0 open-source software. If the data were normally distributed, one-way ANOVA analysis with posthoc Tukey test was applied to identify significant differences between groups of values (relative abundance of taxa, diversity indices, or environmental variables value). If Shapiro–Wilks’s test failed, a non-parametric Kruskal–Wallis test with Bonferroni correction was performed.

Diversity indices were calculated with PAST 4.03. Platform SPADE R online (Species Prediction and Diversity Estimation [37]) was used to calculate community similarity indices (Sørensen, Jaccard, Bray-Curtis) for each variant. The number of bootstrap replications was 100. 

Microsoft Excel was used to calculate mean values and standard deviation of the variables and calculate and depict the relative abundance of the most representative taxa as well as differences between groups of variants.

## 3. Results

### 3.1. Meliniomyces Dominates Ectomycorrhizae on Birch, Oak, and Pine in the Field

For the trees grown in the field, the naturally occurring mycorrhizae were determined. Eleven morphotypes could be distinguished, three on birches, four on oaks, and four on pines (Figure 1 and Appendix A; Appendix A). The most abundant morphotype for all field trees, irrespective of tree species, was formed with ECM fungus *Meliniomyces bicolor* (the morphotype description corresponding to the dark-colored type of fungus *Cenococcum geophilum*). The remaining morphotypes were specific to the three tree species. While birches were associated with the basidiomycete *Lactarius mammosus* in almost equal amounts as *Meliniomyces*, *Inocybe lacera* was less widely distributed. The oak trees, in addition to the most representative fungus *M. bicolor* (with the morphotype here previously described as *Pinirhiza bicolorata*), showed a subdominant morphotype with a smooth hyphal mantle of light brown to beige color without emanating hyphae that resisted molecular approaches for sequencing. The remaining morphotype was formed by *Cortinarius bivelus*. With the third tree species, pine from the field site, the most abundant morphotypes were formed again by *M. bicolor*, while other basidiomycetes present were *Tomentella ellisii* and minor representatives *Rhizopogon mohelensis* and *Tricholoma argyraceum*.

### 3.2. Comparison to Morphotypes Present on the Trees Planted in Pots

An extensively enlarged root system by forming new roots was visible for pot birches and, to a lesser extent, pot oaks. At the same time, the original roots of all pot plants looked dried, and some were even non-viable. Mycorrhiza observed on pot plants was characterized by a wrinkled mantle surface and light brown to reddish color; the more stable abiotic stress factors in pots resulted in less dominance of the dark *Meliniomyces* ectomycorrhizae. Without artificial inoculation, 12 morphotypes were identified: five for birches, three for oaks, and four for pines (Figure 2).

Among them, a member of the genus *Meliniomyces* and subdominant *Inocybe lacera* occurred (Appendix A). These mycorrhiza types, therefore, had increased in abundance from the time of tree transplantation, and *Meliniomyces* had decreased in the abundance of mycorrhizae after the two years in pots.

To mimic a secondary succession where fungal propagules would already be present in sufficient amounts, potted trees were inoculated in addition to the naturally acquired mycorrhizae. Here, the diversity was slightly lower with 10 instead of 12 morphotypes on birches, while again, three morphotypes were observed on oaks and four on pines (Figure 1). *Pisolithus arhizus* was identified on inoculated pot oaks. However, the low numbers do not allow for direct comparison. Hence more statistical approaches were needed.

A higher diversity was confirmed with Shannon diversity (1.03) and Gini–Simpson indices (0.61) for non-inoculated pot oaks and the lowest for field oaks and inoculated pot birches. The Berger–Parker index had the highest values for inoculated pot birches (0.75) and field oaks, suggesting that common morphotypes mainly dominated ectomycorrhizal communities in these variants (Table 2 and Appendix A).

This was supported by a comparison of similarity indices which revealed that the ectomycorrhizal communities among the trees in one variant were similar (Appendix A), with a high Sørensen similarity index (0.833 for field oaks and field and inoculated pot pines) to 1.000 (determined for non-inoculated pot oaks). Similarly, the Bray–Curtis similarity index, which also includes species abundance, had high values from 0.852 for inoculated pot pines to 1.000 calculated for field and inoculated pot birches, pot oaks, and non-inoculated pot pines. Low values of the Jaccard coefficient were calculated for field oaks (0.500) and field pines (0.500), suggesting that these trees had only 50% of common species. For trees in other variants, the Jaccard coefficient ranged from 0.700 to 1.000 representing high similarity within one tree species.

### 3.3. Functional Diversity of Ectomycorrhizal Community

Among all field trees, the most abundant exploration types were contact and short-distance exploration types (Figure 3, Table 3). ECM fungi *Lactarius mammosus* and *Inocybe lacera* observed on field birches, unidentified field oak morphotype O_F_MT2 with smooth hyphal mantle and no rhizomorphs, dark-colored without emanating hyphae field oak morphotype of ECM fungus *Meliniomyces* and field pine morphotype formed by fungus *Tomentella ellisii* contributed to the group of contact exploration type. 

Dark-colored, with voluminous black emanating hyphae of *M. bicolor,* constituted the single morphotype with short-distance exploration type in trees from the field. *Cortinarius* sp., observed on field oaks, formed fans of ramified rhizomorphs and was attributed to medium-distance fringe subtype exploration type. On one field pine, *T. argyraceum* formed white-colored mycorrhiza with a hairy hyphal mantle, extensive emanating hyphae, and interconnected rhizomorphs that was placed in the medium-distance fringe subtype exploration type. Pine was the only tree species on which long-distance exploration type, formed by *R. mohelnensis*, was identified. It showed a coralloid ramification and formed moderately hairy rhizomorphs.

Pot birches had mycorrhiza grouped into contact and short-distance exploration types. The ratio of relative abundance of exploration types in pots differed from that observed from field trees. While for non-inoculated birches, the amount of short-distance exploration type significantly increased to 45.5%, for the inoculated variants, the contact exploration type constituted 95%.

Pot oaks were dominated by contact exploration type of mycorrhiza. The relative abundance of short-distance exploration type was lower for non-inoculated pot variants compared to the field; however, according to the Kruskal–Wallis test, this difference was not significant (Appendix A). The medium-distance exploration type observed in non-inoculated pots was formed by the fungus related to the genus *Meliniomyces*. The relative abundance of this exploration type was significantly higher than on field plants. Inoculation of oaks led to the loss of medium-distance exploration type mycorrhizae. The subdominant short-distance exploration type in inoculated pots was formed by the fungus *Pisolithus arhizus*, characterized by the woolly hyphal mantle with moderately present emanating hyphae.

Non-inoculated pines in pots formed mycorrhiza with grainy hyphal mantle and elongated well-differentiated rhizomorphs and were classified as a long-distance exploration type. Unidentified short-distance exploration type mycorrhiza with dark-colored, voluminous emanating hyphae was rare and observed only on one tree. Two morphotypes formed contact exploration types with a smooth hyphal mantle, mainly dichotomous branching and no emanating hyphae. One was identified to be formed by *Inocybe lacera*. Inoculation of pot pines resulted in the loss of long-distance exploration type mycorrhizae and led to the dominance of contact and short-distance exploration types. The fungal partner of the contact exploration type was represented by *Inocybe lacera*. Short-distance exploration types consisted of three morphotypes of different morphology. The most representative, with short-distance exploration type, was formed by *M. bicolor* and had a morphology different from morphotypes formed by this species on other trees in all variants; this new morphotype exhibited a brown color, woolly hyphal mantle with rare, white emanating hyphae. A morphotype with abundant dark-colored emanating hyphae was observed only on one inoculated tree with ECM fungus identified at the genus level as *Meliniomyces*. The least abundant morphotype was formed by *R. mohelnensis*, which had a woolly hyphal mantle and short emanating hyphae. Additional inoculation of pines did not significantly change the relative abundance of contact and short-distance exploration types compared to trees from the field or non-inoculated pot plants. Overall, additional inoculation of all plants in pots caused the loss of mycorrhiza with rhizomorphs.

### 3.4. Community Correlations to Soil Parameters

Different soil parameters were measured, such as the content of total carbon (TC), total nitrogen (TN), and total phosphorus (TP), as well as soil pH values (Table 4) and the content of metals in the soil (Figure 4).

Canonical correspondence analysis shows a correlation between soil characteristics and the relative abundance of exploration types (Figure 5A). Contact exploration types correlated with the concentration of Al, Cu, Fe, Sr, and U. Higher concentrations of Pb, Sr, Fe, and U, as well as total phosphorus and C/N ratio, resulted in a higher abundance of short-distance exploration types. Medium-distance exploration types correlated with the concentration of Cs and Mn, while the long-distance exploration types were found to be associated with the concentrations of Zn, Ni, Co, Al, Cu, and soil pH. Although a permutation test (N = 999, *p* = 0.444) did not reveal an overall significant association between soil parameters and exploration types, correlation analysis demonstrated significant positive correlations between short-distance exploration types and TC, TN, and C/N ratio, between medium-distance exploration type and the concentration of Cs, and a negative correlation was found between contact exploration types and TN (Appendix A).

*Lactarius mammosus* and *Inocybe lacera* (both described for birches and forming contact exploration type mycorrhiza) positively correlated in the CCA biplot with the concentrations of Fe, Al, and Cu, and negatively correlated with Mn, Ni, Pb, Sr, and TC and C/N ratios (compare Figure 5B). The non-identified oak morphotype O_F_MT2 (contact exploration type) revealed a positive correlation with nutrients and negatively correlated with the concentrations of Al, U, and soil pH value. A positive correlation was also recorded for the pine morphotype formed by *Tomentella ellisii* (contact exploration type) for Mn, Pb, Sr, and U concentrations as well as soil pH; a negative correlation was observed towards Fe and P. Correlation analysis revealed that associations of all field morphotypes with contact exploration and soil characteristics were significant (Table 5). The morphotype formed by *Cortinarius bivelus* (medium-distance exploration type) on oaks had a positive correlation with total nitrogen. *M. bicolor* and *Meliniomyces* sp. (short-distance exploration type) described for oaks did not correlate with soil characteristics. Pine morphotypes formed by *T. argyraceum* and *R. mohelnensis* correlated with the concentrations of Ni, Pb, Sr, Mn, U, and soil pH; however, correlation analysis did not confirm the significance of these correlations.

### 3.5. Plant Inoculation

Tree species showed different performances related to additional inoculation (Table 6, Appendix A). Non-inoculated birches and pines showed higher survival (66.7 and 80%, respectively), whereas non-inoculated oaks did not perform successfully, and only two oaks out of 12 had leaves at the end of the experiment. Inoculation of the rhizosphere with the blend of mycorrhizal fungi resulted in a decrease in birches’ survival rate (14.3%) and an increase in survival for oaks (69.2%). Although the number of inoculated pines at the end of the experiment was low, the overall survival rate after inoculation remained relatively high (60.0%). 

All three species contained different concentrations of metals depending on the variant of the experiment. Overall, additional fungal inoculation did not affect metal content in pot plant biomass. Although significant differences between field variants and one or both pot variants were observed, no trend in metal accumulation in plant aboveground biomass was determined, and different environmental factors might have caused the changes (Appendix A).

## 4. Discussion

### 4.1. The Former Mining Site Is Characterized by Low Ectomycorrhizal Diversity

In this study, field trees of three species of typical trees in temperate forests were investigated. Using young birch, oak, and pine stands that developed naturally at a former mining site, the native mycorrhizal community could be evaluated and compared to re-potted trees from that same site kept under greenhouse conditions for two years. The mycorrhizal assemblies and their environmental functions could be assessed with that approach under more standardized conditions. Nevertheless, mycorrhizal root tips were limited, and this limited taxon sampling might bias the similarity indices, which might be overcome with a probabilistic approach incorporating [37,38]. Our approach could confirm high values of similarity between ectomycorrhizal communities between trees of one species. The trees’ age, succession stage, and contamination level have been described as influencing ectomycorrhizal associations, with metal-polluted soils sharing low diversity and a highly uneven taxa distribution [39,40,41]. Similar trends were observed in the current study. Experiments on the influence of applied heavy metals on mycorrhiza formation confirmed that the increase in metal content reduced, in general, the diversity of mycorrhiza [42,43]. The site’s age seems to be crucial for fungal community diversity [1,44]. Since metal-polluted soils generally contain a lower number of vital fungal spores and a low density of fine roots is typical for younger trees, a decreased community diversity might be expected [45].

### 4.2. Potential for Afforestation Programs

The success of vegetation establishment depends on adaptation to the newly created conditions in post-mining landscapes [46]. The substrate generally is spacially heterogenous, contains low amounts of nutrients, and is prone to have a low pH, increasing the mobility of heavy metals with capillary transport of water. Even if (a limited layer of) allochthonous soil cover is applied, this will not provide the water holding capacity nor the abundance of ectomycorrhizal fungal propagules present in native sites. However, ectomycorrhiza can be beneficial for plant establishment, specifically under detrimental conditions. Fertilization will alleviate the scarcity of nutrients but, at the same time, is known to suppress the formation of sustainable mycorrhiza. Hence, we investigated the addition of fungal inoculum that might provide a measure easily applicable in such afforestation programs on former mining sites.

Trees to be planted should be able to compete and establish a viable forest ecosystem. Among the typical woody pioneer species which can naturally colonize post-mining areas are birches [47]. While they facilitate soil functioning and biodiversity, birches are known to be particularly sensitive to intraspecific competition [48]. In afforestation, plant biomass should be evaluated for the uptake of metals. Here again, mycorrhizae provide a barrier with the hyphal mantle protecting root tips from excess heavy metals, and metal accumulation at and inside the hyphae without transfer to the host tree can protect the trees from toxic element concentrations under field conditions. In our analysis, we, therefore, checked our pot birches and found that they accumulated only Mn and Cd, while in the field, Fe, Cu, Sr, Pb, and U uptake was higher. Still, the measured concentrations are below phytotoxicity levels. Thus, (mycorrhizal) birches can be considered phytostabilizing as the biomass does not contain levels exceeding threshold concentrations. Hence, in addition to providing a positive effect on stabilizing the substrate from erosion and allowing for recreational use, birch wood from mining site reforestation provides a potential crop for wood harvesting with minimal risk of soil re-contamination through leaf fall.

Oak, in contrast to birch, is attributed to climax or late successional-stage species in the European temperate forests [49]. At Kanigsberg, sites with better soil quality and nutrient contents would be specifically prone to harbor oaks known to accumulate heavy metals in their biomass [49]. After transfer in pots, oaks accumulated Mn, potentially related to low P contents, as plants are known to exude more organic acids to obtain P, which facilitates metal uptake, specifically under moist conditions [50]. This is accompanied by the observed high mortality in non-inoculated pots, while mycorrhiza could stabilize oak performance showing that the implementation of mycorrhizal blends might considerably improve the growth of oaks in post-mining areas.

Pines represent typical pioneer trees widely used in reclamation [46,51,52]. Fertilization improved their growth in polluted soils. However, that may come at the cost of phytoextraction, visible in both field and pot pines containing elevated Al [53]. Here, the pot experiment showed that pines were the least affected by disturbance, regardless if they were inoculated with mycorrhizal fungi. Transfer in pots led to the accumulation of Mn, Co, and Cu into the needles, while Cr, Fe, Ni, and Sr were excluded. Overall, as the turnover of aboveground biomass in coniferous tree stands is markedly slower than in deciduous tree stands, coniferous trees are a good alternative for the reclamation of areas contaminated with heavy metals.

### 4.3. Tree Transplanting Affected Functional Diversity

Transfer of the plants from the field to pots affected the tree species differently. The low survival rate of non-inoculated variants (lowest for oak with 16.7%) was correlated with an enlarged root system by forming new roots for birches, less for oaks, and remained unchanged for pines, consistent with the trees’ ability to perform well in primary, early succession. In non-inoculated oaks and pines, an abundance of ectomycorrhizae with rhizomorphs increased significantly. The development of rhizomorphs could facilitate the extensive exploration of newly created niches.

Besides, the hydrophobicity of rhizomorphs can indirectly control the access of water into mycelium. Therefore, for oaks and pines, the development of hydrophobic rhizomorphs might be a strategy for improving effective nutrient transport, as hydrophobic properties will likely prevent the loss of solutes during transfer to the host plant [16]. The application of the ectomycorrhizal fungi by inoculation generally could contribute to the improvement of the water regime but might lead to the loss of rhizomorphs as more carbon-costly and unnecessary in pot conditions.

### 4.4. Relationships to Ecosystem Functioning and Succession

Among the field birches, *Lactarius* was abundant, potentially related to the high tolerance to unfavorable local conditions, particularly high aluminum contents [54,55]. For both *Cortinarius* and *Tricholoma*, extensive hydrophobic rhizomorphs have been reported at contaminated sites [16,41,56] that enhance the acquisition of nitrogen from organic sources [57]. *Rhizopogon mohelnensis* would be typically found in early, primary succession [58,59]. *Thelephoraceae* with *Tomentella ellisii* and *Meliniomyces* have been described commonly in coniferous forests and are often found in contaminated soils [60,61].

Field morphotypes formed mycorrhiza preferentially with contact and short-distance exploration types usually described for undisturbed forest stands [62]. However, the exploration types in metal-affected soils consisted of increasing numbers of morphotypes with short and medium-distance exploration types [40]. It has been discussed that abundant emanating hyphae and ramifying rhizomorphs typical for medium-distance exploration types can function as a filter hindering the entry of heavy metals into the host plant cells. At the same time, they can explore a larger soil volume for nutrient acquisition, which is even better for long-distance exploration types that are usually described for trees in metal-contaminated areas [56]. Here, permutation analysis did not reveal a significant correspondence between metals and exploration types, leading to the conclusion that abiotic parameters of the disturbed and coarse substrate might override the intricate relationships between exploration types and heterogeneous metal bioavailability. Only pines formed mycorrhiza with woolly silver mantle with infrequent emanating hyphae and no rhizomorphs with *Rhizopogon mohelnensis*. This species also was detected on field pine roots, where it formed distinctive rhizomorphs. Again, hydrophobic mycelium and rhizomorphs seem to indicate rather dry conditions [63,64].

With respect to succession, the spatial position of short roots might influence the choice of the mycorrhizal partner fungus. Bruns [58] argued that short roots close to the stem accumulate more carbohydrates, supporting infection by carbon-demanding, late-stage fungal symbionts. At the same time, these are prone to produce rhizomorphs, as the center of the plant root is deprived of mineral nutrients. Since we selected healthy short roots found more often at a distance from the stem, the study might carry an intrinsic bias towards early-stage mycorrhizal fungi.

Overall, it can be concluded that field plants promoted ecological filtering toward selecting specific symbiotic fungi with specific exploration types, which would contribute to plants’ tolerance to abiotic conditions specific at each sampling site.

## 5. Conclusions

With our approach, we could show that even on substrates in post-mining landscapes, afforestation is possible, and the trees can establish a viable root system. This will be essential to stabilize the soil and protect it from water and wind erosion. At the same time, the presence of mycorrhizal fungi can positively affect water availability. While the young trees did not yet develop high amounts of long-distance exploration type mycorrhizae, still a succession from constant to medium-distance exploration types were found, showing that establishing sustainable ecosystems even under these detrimental conditions is possible. The addition of mycorrhizal fungal inoculum could help to facilitate this process of increasing mycorrhization that, in the end, helps the trees survive the planting. The use of a mixed forest consisting of deciduous birches and oaks, and coniferous pines seems well suited to provide not only a sustainable land use for society’s recreational activities but the phytostabilization features observed will also allow for forest harvesting of these newly established forests in future generations.

## Figures and Tables

**Figure 1 jof-09-00483-f001:**
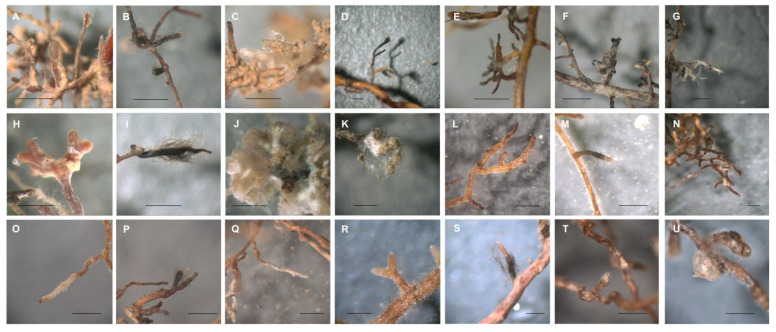
Morphotypes observed on field plants. Birch roots with (**A**) *Lactarius mammosus*, (**B**) *Meliniomyces bicolor* (*Pinirhiza bicolorata*-type with emanating hyphae), and (**C**) *Inocybe lacera*; oak roots with (**D**) *Meliniomyces bicolor* (*Cenococcum geophilum*-type), (**E**) non-identified O_F_MT2, (**F**) *Meliniomyces bicolor* (*Pinirhiza bicolorata*-type w/o emanating hyphae), and (**G**) *Cortinarius bivelus*; pine roots with (**H**) *Tomentella ellisii*, (**I**) *Meliniomyces bicolor* (*C. geophilum*-type), (**J**) *Rhizopogon mohelnensis*, and (**K**) *Tricholoma argyraceum*. With inoculated pot plants, birch roots formed mycorrhizae with (**L**) non-identified B_inoc_MT1, (**M**) non-identified B_inoc_MT2, and (**N**) non-identified B_inoc_MT3; oaks roots with (**O**) non-identified O_inoc_MT1, (**P**) non-identified O_inoc_MT2, (**Q**) *Pisolithus arhizus*; and pine roots with (**R**) *Inocybe lacera*, (**S**) *Meliniomyces* sp., (**T**) *Meliniomyces bicolor*, and (**U**) *Rhizopogon mohelnensis*. Bar always represents 1.0 mm.

**Figure 2 jof-09-00483-f002:**
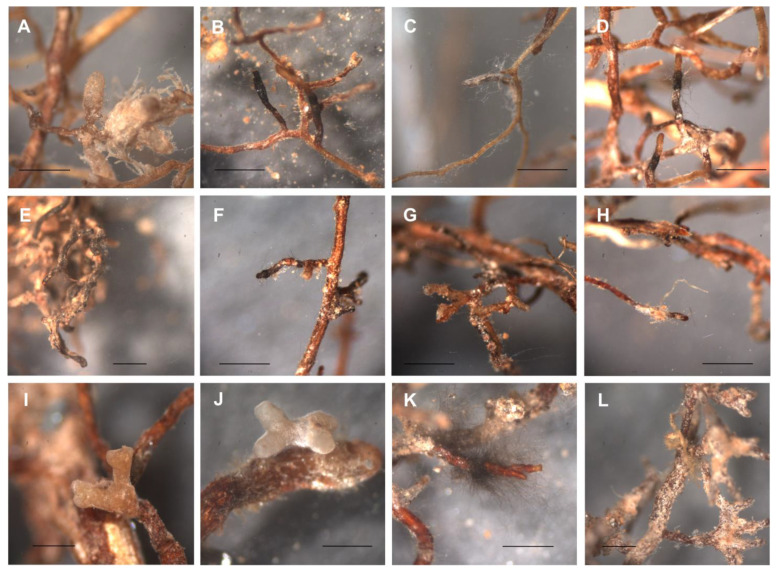
Morphotypes observed on non-inoculated pot plants. Birch roots with (**A**) non-identified B_non_inoc_MT1, (**B**) non-identified B_non_inoc_MT2, (**C**) non-identified B_non_inoc_MT3, (**D**) non-identified B_non_inoc_MT4, (**E**) non-identified B_non_inoc_MT5; oak roots with (**F**) non-identified O_non_inoc_MT1, (**G**) non-identified O_non_inoc_MT2, (**H**) *Meliniomyces*; pine roots with (**I**) *Inocybe lacera*, (**J**) non-identified P_non_inoc_MT2, (**K**) non-identified P_non_inoc_MT3, (**L**) non-identified P_non_inoc_MT4. Bar always represents 1.0 mm.

**Figure 3 jof-09-00483-f003:**
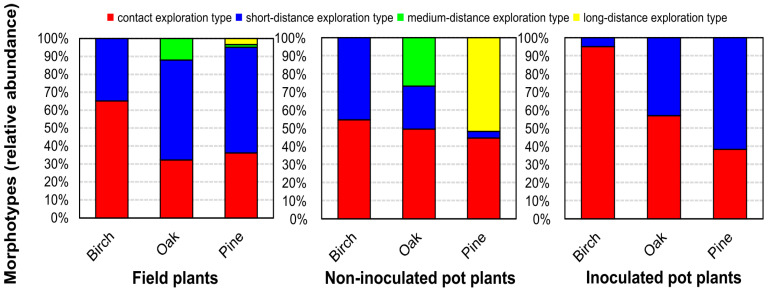
Relative abundance of exploration types of field plants, non-inoculated pot plants, and inoculated pot plants.

**Figure 4 jof-09-00483-f004:**
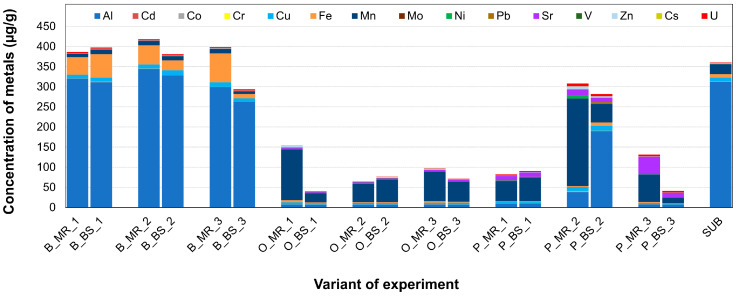
Sum content of toxic metals’ bioavailable fractions determined in test field soil. B—birch, O—oak, P—pine; MR—mycorrhizosphere of field plant, BS—bulk soil, SUB—control pot substrate.

**Figure 5 jof-09-00483-f005:**
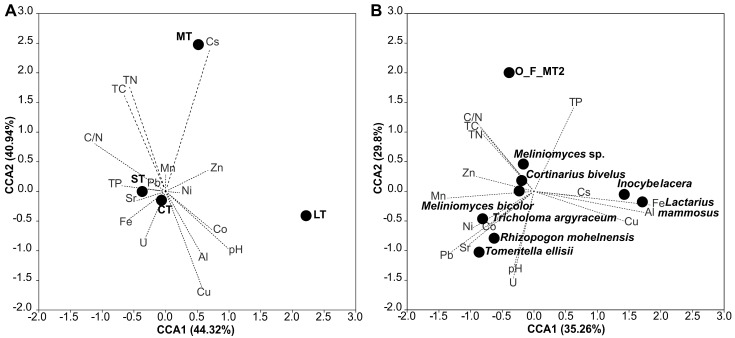
Canonical correspondence analysis biplot representing the correlation (**A**) between soil parameters and exploration types of mycorrhizae and (**B**) between soil parameters and mycorrhizal taxa described for field plants. CT—contact exploration type, ST—short-distance exploration type, MT—medium-distance exploration type, LT—long-distance exploration type.

**Table 1 jof-09-00483-t001:** Field sampling sites.

Sampling Site	Location
Birch and soil sampling site	50°49′35.35″ N, 12°9′11.68″ E
Oak sampling site	50°49′36.13″ N, 12°9′12.56″ E
Pine sampling site	50°49′39.65″ N, 12°9′17.87″ E

**Table 2 jof-09-00483-t002:** Diversity indices calculated for ectomycorrhizal fungi communities in different variants of the experiment.

Variant	H_SD_	H_GS_	H_SH_	H_BP_
Field birches	0.60 ± 0.11	0.39 ± 0.11	0.64 ± 0.16	0.72 ± 0.13
Field oaks	0.66 ± 0.20	0.34 ± 0.20	0.55 ± 0.30	0.73 ± 0.21
Field pines	0.55 ± 0.14	0.45 ± 0.14	0.71 ± 0.24	0.66 ± 0.15
Non-inoculated pot birches	0.48 ± 0.05	0.52 ± 0.05	0.90 ± 0.13	0.60 ± 0.07
Non-inoculated pot oaks	0.39 ± 0.06	0.61 ± 0.06	1.03 ± 0.08	0.50 ± 0.13
Non-inoculated pot pines	0.57 ± 0.13	0.43 ± 0.13	0.77 ± 0.16	0.69 ± 0.16
Inoculated pot birches	0.68 ± 0.36	0.32 ± 0.36	0.56 ± 0.55	0.75 ± 0.31
Inoculated pot oaks	0.51 ± 0.11	0.49 ± 0.11	0.77 ± 0.18	0.59 ± 0.14
Inoculated pot pines	0.49 ± 0.13	0.51 ± 0.17	0.82 ± 0.21	0.57 ± 0.17

H_SH_—Shannon diversity index, H_GS_—Gini–Simpson index, H_SD_—Simpson dominance index, H_BP_—Berger–Parker index.

**Table 3 jof-09-00483-t003:** Summarized data of morphotypes, the identification of ECM fungus, and exploration type.

Host	Variant	Morphotype	Molecular Identification	Exploration Type
Birch	Field plant	B_F_MT1	*Lactarius mammosus*	contact
		B_F_MT2	*Meliniomyces bicolor*	short
		B_F_MT3	*Inocybe lacera*	contact
	Pot non-inoculated plant	B_non_inoc_MT1	non-identified	contact
		B_non_inoc_MT2	non-identified	short
		B_non_inoc_MT3	non-identified	short
		B_non_inoc_MT4	non-identified	short
		B_non_inoc_MT5	non-identified	short
	Pot inoculated plant	B_inoc_MT1	non-identified	contact
		B_inoc_MT2	non-identified	short
		B_inoc_MT3	non-identified	contact
Oak	Field plant	O_F_MT1	*Meliniomyces bicolor*	short
		O_F_MT2	non-identified	contact
		O_F_MT3	*Meliniomyces*	contact
		O_F_MT4	*Cortinarius bivelus*	medium, fringe
	Pot non-inoculated plant	O_non_inoc_MT1	non-identified	short
		O_non_inoc_MT2	non-identified	contact
		O_non_inoc_MT3	*Meliniomyces*	medium, smooth subtype
	Pot inoculated plant	O_inoc_MT1	non-identified	contact
		O_inoc_MT2	non-identified	short
		O_inoc_MT3	*Pisolithus arhizus*	short
Pine	Field plant	P_F_MT1	*Tomentella ellisii*	contact
		P_F_MT2	*Meliniomyces bicolor*	short
		P_F_MT3	*Rhizopogon mohelnensis*	medium, fringe
		P_F_MT4	*Tricholoma argyraceum*	medium, mat
	Pot non-inoculated plant	P_non_inoc_MT1	*Meliniomyces*	contact
		P_non_inoc_MT2	non-identified	contact
		P_non_inoc_MT3	non-identified	short
		P_non_inoc_MT4	non-identified	long
	Pot inoculated plant	P_inoc_MT1	*Inocybe lacera*	contact
		P_inoc_MT2	*Meliniomyces*	short
		P_inoc_MT3	*Meliniomyces bicolor*	short
		P_inoc_MT4	*Rhizopogon mohelnensis*	short

**Table 4 jof-09-00483-t004:** Selected soil chemical parameters.

Variant of Experiment	pH	TC (%)	TN (%)	TP (mg/kg)
Birch_mycorrhizosphere	4.44 ± 0.32	0.87 ± 0.24	0.11 ± 0.01	669 ± 173
Birch_bulk soil	3.75 ± 0.30	1.07 ± 0.05	0.13 ± 0.01	603 ± 198
Birch_pot substrate	6.03 ± 0.36	0.39 ± 0.05	0.07 ± 0.00	769 ± 285
Oak_mycorrhizosphere	3.54 ± 0.10	3.45 ± 0.35	0.31 ± 0.05	786 ± 127
Oak bulk soil	3.56 ± 0.09	2.60 ± 0.85	0.27 ± 0.07	802 ± 193
Oak pot substrate	5.32 ± 0.42	0.36 ± 0.01	0.07 ± 0.00	843 ± 100
Pine_mycorrhizosphere	5.20 ± 1.47	0.87 ± 0.13	0.09 ± 0.01	347 ± 61
Pine_bulk soil	6.21 ± 0.53	1.11 ± 0.44	0.10 ± 0.03	369 ± 37
Pine_pot substrate	5.52 ± 0.22	0.37 ± 0.01	0.07 ± 0.00	718 ± 178
Control pot substrate	3.47 ± 0.01	0.33 ± 0.01	0.07 ± 0.00	801 ± 121

TC, TN, TP—total carbon, nitrogen, phosphorus.

**Table 5 jof-09-00483-t005:** Coefficients of correlation between soil characteristics and ectomycorrhizal morphotypes and ECM fungi described for the field plants.

Soil Characteristics	*Lactarius mammosus*	*M. bicolor*	*Inocybe lacera*	O_F_MT2	*Meliniomyces*	*Cortinarius bivelus*	*Tomentella ellisii*	*R. mohelnensis*	*T. argyraceum*
Al	**0.76 ***	−0.36	**0.79 ***	**−0.69 ***	−0.44	−0.35	0.11	0.17	0.10
Co	−0.44	0.02	−0.40	0.19	0.26	0.21	0.26	0.45	0.38
Cu	**0.75 ***	−0.49	**0.75 ***	−0.42	−0.37	−0.41	−0.13	0.24	0.10
Fe	**0.79 ***	−0.37	**0.78 ***	−0.03	−0.13	0.10	**−0.65 ***	−0.38	−0.17
Mn	**−0.77 ***	0.11	**−0.77 ***	0.15	−0.04	0.20	**0.57 ***	0.45	0.38
Ni	**−0.78 ***	0.23	**−0.75 ***	0.20	0.25	0.10	0.53	0.45	0.38
Pb	**−0.78 ***	0.21	**−0.78 ***	−0.07	−0.04	0.06	**0.79 ***	0.45	0.28
Sr	**−0.76 ***	0.17	**−0.76 ***	−0.06	−0.09	0.00	**0.84 ***	0.24	0.38
Zn	−0.48	−0.11	−0.49	0.36	0.14	0.31	0.12	0.45	0.31
Cs	0.44	−0.02	0.45	−0.04	0.09	0.43	−0.52	0.00	−0.14
U	−0.02	−0.01	−0.01	**−0.67 ***	−0.36	−0.50	**0.82 ***	0.45	0.38
TC	**−0.57 ***	0.34	**−0.54 ***	**0.61 ***	0.52	0.47	−0.20	−0.28	−0.03
TN	**−0.54 ***	0.33	−0.50	**0.54 ***	0.42	**0.54 ***	−0.22	−0.24	−0.07
C/N	**−0.61 ***	0.35	**−0.61 ***	**0.69 ***	0.44	0.35	−0.24	−0.38	−0.17
TP	0.17	0.03	0.22	**0.58 ***	0.28	0.14	**−0.84 ***	−0.24	−0.38
pH	0.14	−0.18	0.14	**−0.64 ***	−0.39	−0.27	**0.65 ***	0.45	0.38

Asterisks and bold print represent a significant correlation (*p* < 0.05). TC, TN, TP—total carbon, nitrogen, phosphorus; O_F_MT2: non-identified.

**Table 6 jof-09-00483-t006:** Pot plant’s survival rates.

Tree Species		Total	Plant Survival Rate (%)
Birch	Number of plants non-inoculated	12	66.7
	Number of plants inoculated	14	14.3
Oak	Number of plants non-inoculated	12	16.7
	Number of plants inoculated	14	69.2
Pine	Number of plants non-inoculated	10	80.0
	Number of plants inoculated	10	60.0

## Data Availability

All data are provided with this submission.

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
