# Peer review of "Ectomycorrhizal Community Shifts at a Former Uranium Mining Site"

_jof, 2023, doi:10.3390/jof9040483_

Round 1

Reviewer 1 Report

The manuscript is very interesting and I would like to do my compliments to the Authors for the great amount of work done. However, that make the manuscript little confusing.

I really appreciate the morphotyping and the molecular identification of each morphotype. However, in the test there is a great confusion. In some parts of the results are reported the morphotypes acronyms  (for example line 317) in other pats the morphotype names (line 182) and in several parts more correctly the molecularly identified OTUs. I suggest to use in all the test the identified fungal OTU and to indicate only the morphotype when molecular identification was unsuccessful. You can also indicate between brackets the morphotype name after the OTU when it is necessary.

It is not so clear why the Authors did the pot experiments and above all why they inoculate the trees in the pots. In the pots the conditions are completely different from the field, moreover in my opinion, inoculation do not mimic secondary succession of fungi in ectomycorrhizal communities. Several factors including  plant ages and soil physic-chemical changes  determine the secondary succession in ECM fungal communities. Inoculation with late stage mycorrhizal fungi in young seedlings could lead a partial failure such as in the presented work.

Moreover, the discussion and the first part of the conclusions are not clear.

Tables in the text have problems:

Table 1 the columns 1 and 3 are identical

Table 3  only the columns 1,2 and 5 are informative. Figure 5 is a repetition of table 3 and it can be included as supplemental material.

The soil analyses are lacking also in the supplemental materials.

I suggest to the Authors to revise simplifying the manuscript and to resubmit it.

Author Response

Thank you for your support, see uploaded document for PointbyPoints, please.

Reviewer 2 Report

This manuscript describes ectomycorrhizal community at uranium mining sites. It is an interesting study, in my opinion, well-publishable in JoF.

I have several comments and questions listed below.

1) The whole text: Scientific names at all taxonomic ranks should be written in italics according to Thines et al.: Thines, M., Aoki, T., Crous, P.W. et al. Setting scientific names at all taxonomic ranks in italics facilitates their quick recognition in scientific papers. IMA Fungus 11, 25 (2020). https://doi.org/10.1186/s43008-020-00048-6

2) Line 17: do you mean soil contents?

3) Line 106: Hebeloma crustuliniforme, not "crustiliniforme"

4) 2.2: Analytical control is missing. Were certified reference materials measured?

5) 2.2: The sequential extraction was conducted according to an obscure paper (sorry) published in German and not available on the internet. The fractions I and II are not defined in the paper. Why this extraction scheme was used (and not, e.g., well-known Tessier scheme or BCR?). In any case, define the fractions and provide information how the extraction was conducted. Providing the reference is not enough in this case.

Furthermore, analytical data is not presented in the Supplementary material. The reader cannot recognize if the soils can be considered polluted/contamined or not. This information should be included in the Supplementary.

6) L. 179-180: I wonder why Lactarius sp. and Mallocybe sp. could not be determined at species level? I see the comparison in the supplementary but if there is a match at species level, it should be possible to identify the species and present them in the text or not?

In the Supplementary, you present "B_F_MT3" as Mallocybe sp. (Unite) and Inocybe lacera (GenBank). But Inocybe lacera is not a Mallocybe species! So whas it Inocybe or Mallocybe?

I would expect you to submit all your seqences to GenBank and compare them with the databe(s) + thoroughly interpret the data at species level. Ditte Bandini could probably help you with Inocybe s.l.? (https://www.inocybe.org/)

7) L. 186: italics to be used in one name

8) L. 243: italics to be used in names

9) L. 305 total C and total N should be abbreviated as indicated above in the text

Question 1: was there any logic in the selection of elements? Why, e.g., Cd, Rb or Th are missing?

Question 2: You claim "Pines represent typical pioneer trees widely used in reclamation [37,42,43]. Fertilization improved their growth in polluted soils. However, that may come at cost of phytoextraction, visible in both field and pot pines containing elevated Al [44]. Regardless of inoculation, pines excluded Cr, Fe, Ni and Sr making the coniferous tree a good alternative for remediation of soils contaminated with heavy metals."

For phytoremediation, I would expect using trees accumulating metals, not rejecting. Can you explain this?

Suggestions (optional): In the introduction, it would be worth mentioning an old study dealing with ECM fungi at uranium tailings (https://doi.org/10.1016/0048-9697(81)90120-0). Furthermore, studies on ectomycorrhizae from polluted areas, even from uranium mining areas, have been published, e.g. https://doi.org/10.1016/j.jhazmat.2014.07.050. What do you think about the theory that mushrooms form a barrier to metal toxicity via the ectomycorrhizal mantles? (10.1007/s10646-007-0149-x, https://doi.org/10.1016/j.envpol.2016.08.009). Would that be relevant for symbionts at your sites?

Author Response

(The authors gave the same response as above.)

Reviewer 3 Report

Review: Ectomycorrhizal community shifts at a former uranium mining site

This is an interesting project!  And a big one – I respect the amount of work and data that the authors have completed.  I don’t think it is ready for publication and hope and encourage the authors to revise and resubmit. 

The main issues by section:

Introduction: needs more literature review on ECM in soils with heavy metals

Methods: incomplete and needs much more detail

Results: reads like authors are still considering how to format this and what to include and supplement

Discussion: reads like the authors are still synthesizing and making connections in the discussion. 

My line items follow:

Introduction:

Line 28 edit to: At the same time, the substate is characterized by low porosity, decreased water-holding capacity, and the lack of water-stable aggregates due to compaction.

Line 40 edit to: Therefore, the sustainability and stability of land-cover needs improvement especially during heavy rain events or drought spells that are becoming more prominent with climate change.

Line 49: author mentions “heavy metals”, however, more introduction needs to focus here.  What is known about ECM and mined systems? Fungi and metals? 

Line 57: clarify “exchange of signals”

Lines 61-64: this is an important part of the introduction.  Break this sentence up and expand on what is known about fungi and disturbed sites.

Lines 65-69:  connect this with rhizomorph formation.  Also, this may relate to disturbed sites, but how does this relate to sites with heavy metals?  Also – what is the timeline the author is referring to regarding fungal succession?  Is Cortinarius truly a later successional stage fungus? 

Lines 70-72: briefly discuss this, it is key to your objective. 

Lines 72-74: Expand here: where are the metals being accumulated? Plant?  Fungi? Both?

Lines 77-82: begin to clearly solidify you research question regarding rhizomorph and fungal succession.

Methods:

Line 86: what metals are being targeted? What was the pH?

Line 87: how many samples? Were the plants planted or did they come in naturally?

Line 90: what depth?

Line 101: where did the trees come from? Did they have ECM at time of planting?

Lines 139-141: how were the rhizomorphs characterized?  Where they quantified in any way?

151-153: Did you use two methods of DNA extraction/PCR? Why?

Line 166:  How similar where your samples to the BLAST and UNITE databases?  Did you use a cut-off?  For example, sequences 96% similarity or greater were used in ID?

Line 168: Data Processing.  This section is incomplete.  What were the factors/ Responses?  Program and package used?  Other stats referenced throughout the paper are missing in this section. 

Results:

Fig 1 and 2 – nice photos!  No photos available for the inoculated trees?  It is ok, just curious why?

Line 217: substitute inoculated for inoculation

Line 220 – delete “Interestingly”

Lines 223-227 – How where these differences confirmed?  I did not see the analysis in the methods. 

Table 2: needs some type of statistics.  Sub-script at bottom of table is unclear.

Line 243 Italicize species names

Figure 3: help reader by adding main titles to each plot and a color key that indicates exploration type

Line 268: Kristal Wallis not in methods

Around the paragraph (lines 275-285) help reader by having a table.  Something that combines tables S3 and s4 and provides the reader with a summary of exploration types and accession numbers.  Did you deposit your sequences?  If so, provide you accession #s.

Line 296: CCA “could” show a correlation?  Unclear.

Lines 302-308: this needs to be broken up because it is really difficult to read and interpret.  Figure 4A really does not help.  Consider deleting this graph and adding Table S8.    

Figure 4B is interesting – make larger so reader can interpret.

Lines 321-325: consider adding Table S9 to help reader.  Interesting…

Line 324: was that total soil nitrogen?

Table 3. Statistics?

Lines 340-341:  is this significant?  Did you use statistics to make this conclusion? 

Figure 5: needs to show variability – error bars or confidence intervals.  What was used in this analysis? 

Lines 345-349: Interesting – did fungal species influence metal uptake?   

Discussion:

Lines 353-355: Clarify and expand on what you are saying here.

360-364: Unclear statements. 

At this point – the discussion is starting to read very awkwardly.  I can appreciate the amount of information that the author is synthesizing – but it reads very unclearly

Lines 367-370: not needed. 

Lines 371-374: these are important points but in a run on sentence. 

Lines 385 -386 not sure if the data show this – double check. 

Lines 386-387: pines excluded some metals – but what about the others? 

In general: section 4.2 reads a bit disconnected to the data that was presented.  incomplete and never really connects back to the importance of the paper. 

Lines 390-393: awkward and not connecting to the data presented.

Lines 397-401: important… but missing a connection.

Author Response

(The authors gave the same response as above.)

Round 2

Reviewer 1 Report

The manuscript is considerably improved and now it is clear why the Authors inoculated the plants in the pots.

However,  in some few points there is still confusion between morphotypes and ECM fungi.  A morphotype is formed by a fungus is not a fungus (see for example abstract line 8, and manuscript lines 251, 252, 256).

Line 188- 189. This is true but it is not a method.

Line 262 delete “a member of”

Lines 464-473, 583-585, 1198-1206,  In the results the plant mycorrhization rates are not reported. The diversity of ECM fungal communities is analyzed. Please modify these sentences according your results.

Author Response

Thank you very much for your helpful comments, see details in the attached document with Point by points.

Reviewer 2 Report

The authors have gone to great lengths to revise the text. I think the manuscript has been much improved, congratulations. I recommend publishing the revised form in JoF.

When checking the proof, please correct the typos I noted in the revision:

Line 71: Correct "Incybe" to Inocybe

Line 542: Correct "mohlensis" to "mohelnensis"

Author Response

Thank you very much for your helpful guidance, see details in the attached document with Point by points.
